# Severe hypoxia drives loss of ST6GAL1-mediated α2,6-sialylation in the epicardial secretome impairing angiogenic activity

Junqing Zhang[1], Harry J. Whitwell[2,3], Costanza Emanueli[1] and Pilar Ruiz-Lozano[1,4,*]

## ABSTRACT

The epicardium contributes to cardiac development and regeneration, primarily through paracrine signalling. However, the impact of severe hypoxic stress, such as occurs immediately following myocardial infarction, on the activity of the epicardial secretome remains poorly defined. Here we investigated the angiogenic potential of the epicardial secretome under normoxic and hypoxic conditions. In contrast to expectation, the results demonstrate that the angiogenic effect of epicardial cells on vascular endothelium is hampered under severe hypoxia. Mechanistically, we found that hypoxia-mediated sialylation remodels the epicardial secretome altering its angiogenic potential. Severe hypoxia suppresses *St6gal1* gene expression decreasing the level of α2,6-sialylation, as measured by comprehensive lectin profiling and sialyltransferase analysis. Functional studies revealed that both global and α2,6-specific sialylation inhibition markedly compromised the pro-angiogenic capacity of the epicardial secretome, reducing the *in vitro* tube formation. Our findings indicate that ST6GAL1-dependent α2,6-sialylation is required to activate the pro-angiogenic potential of the epicardial secretome. These data also suggest that glycosylation is a modifiable response to ischemic injury, offering a new therapeutic target for enhancing tissue repair.

KEY WORDS: Epicardium, Secretome, Hypoxia, Angiogenesis, Sialylation

## INTRODUCTION

The mesothelial layer enveloping the heart, the epicardium, plays a pivotal role in cardiac development by undergoing epithelial-to-mesenchymal transition (EMT) and multipotent differentiation (Quijada et al., 2020). Studies in zebrafish have demonstrated that epicardial cells orchestrate cardiac repair (Lepilina et al., 2006), and that epicardial paracrine signals are required for blood vessel formation in the regenerating fish (Kim et al., 2010; Kikuchi et al., 2011).

Although gene expression studies in the adult mammalian heart support the idea that ischemic injury activates the angiogenic potential of the epicardial secretome (Zhou et al., 2011), functional evidence of induced angiogenic activity is lacking. The embryonic mammalian epicardium produces a wide array of angiogenic factors that support fetal heart angiogenesis and myocardial development, including fibroblast growth factors (FGFs), platelet-derived growth factors (PDGFs), insulin-like growth factors (IGFs), vascular endothelial growth factors (VEGFs), and extracellular matrix (ECM) components (Wong et al., 2024; Zhou et al., 2011), but whether severe hypoxia in the adult reactivates this pro-angiogenic program remains unclear.

Here, we sought to analyse whether changes in oxygen tension influence the activity of epicardial secreted proteins. Oxygen tension can drop below 1% in the infarct zone, creating an extremely hypoxic niche characterized by abrupt ATP depletion, metabolic acidosis, and inflammatory activation (Jennings and Reimer, 1991; Zhu et al., 2025), which is fundamentally distinct from the milder, physiological hypoxia (∼2-5% $O_2$) observed during development or later-stage repair (Bellio et al., 2016; Giaccia et al., 2004; Hadjipanayi et al., 2010). While hypoxia broadly affects the cellular secretome (Cosme et al., 2017; Kukumberg et al., 2021; Enström et al., 2024), its effect on paracrine signalling remains unexplored. This gap is particularly significant given the crucial role of post-translational modifications (PTMs) in the regulation secreted protein bioactivity (McCaffrey and Braakman, 2016; Wu and Krijgsveld, 2024). Glycosylation, as one of the most diverse PTMs, modulates protein half-life, ligand-receptor interactions, and ECM remodelling (Schjoldager et al., 2020; Loaeza-Reyes et al., 2021; He et al., 2024; Chatham and Patel, 2024). Thus, while moderate hypoxia promotes angiogenesis, it is unclear whether extreme, acute hypoxia compromises epicardial reparative signalling.

In this study, we investigated how hypoxia reprograms glycosylation of the epicardial secretome and explored the functional consequences for cardiac angiogenesis. We found that severe hypoxia, in contrast to moderate hypoxia, impaired glycosylation and pro-angiogenic potential of the secretome. We went on to characterize the glycan modifications and glycosyl transferases responsible and found that terminal sialylation is a regulated step in the angiogenic response to hypoxia. The results have implications for repair of cardiac and potentially other tissues following severe ischemic injury.

## RESULTS

### Hypoxic stress impairs the pro-angiogenic potential of epicardial secretome

To investigate changes in the epicardial secretome under hypoxia, we employed a well-controlled *in vitro* model system using primary rat EMCs (cell characterization shown in Fig. S1). EMCs were exposed to three precise-regulated oxygen conditions: (i) normoxia (NM, 20% ambient $O_2$ for 48 h), (ii) hypoxia (HP, 0.1% $O_2$ for 48 h), and (iii) reoxygenation (RO, 0.1% $O_2$ for 24 h followed by 20% $O_2$ for 24 h) (Fig. 1A). Oxygen level in EMCs was monitored using the Image-iT[TM] hypoxia reagent (Fig. 1B), with quantitative fluorescence intensity measurements obtained at 24 and 48 h (Fig. S2A). At 24 h,

[1]National Heart and Lung Institute, Imperial College London, London W12 0NN, United Kingdom. [2]Section of Bioanalytical Chemistry, Division of Systems Medicine, Faculty of Medicine, Imperial College London, London W12 0NN, United Kingdom. [3]National Phenome Centre and Imperial Clinical Phenotyping Centre, Department of Metabolism, Digestion and Reproduction, Imperial College London, London W12 0NN, United Kingdom. [4]Regencor, Inc, California 94070, USA.

*Author for correspondence (pilar@regencor.com)

J.Z., 0000-0001-5283-1648; H.J.W., 0000-0001-8987-4158; C.E., 0000-0002-2392-0702; P.R.-L., 0000-0002-7150-4615

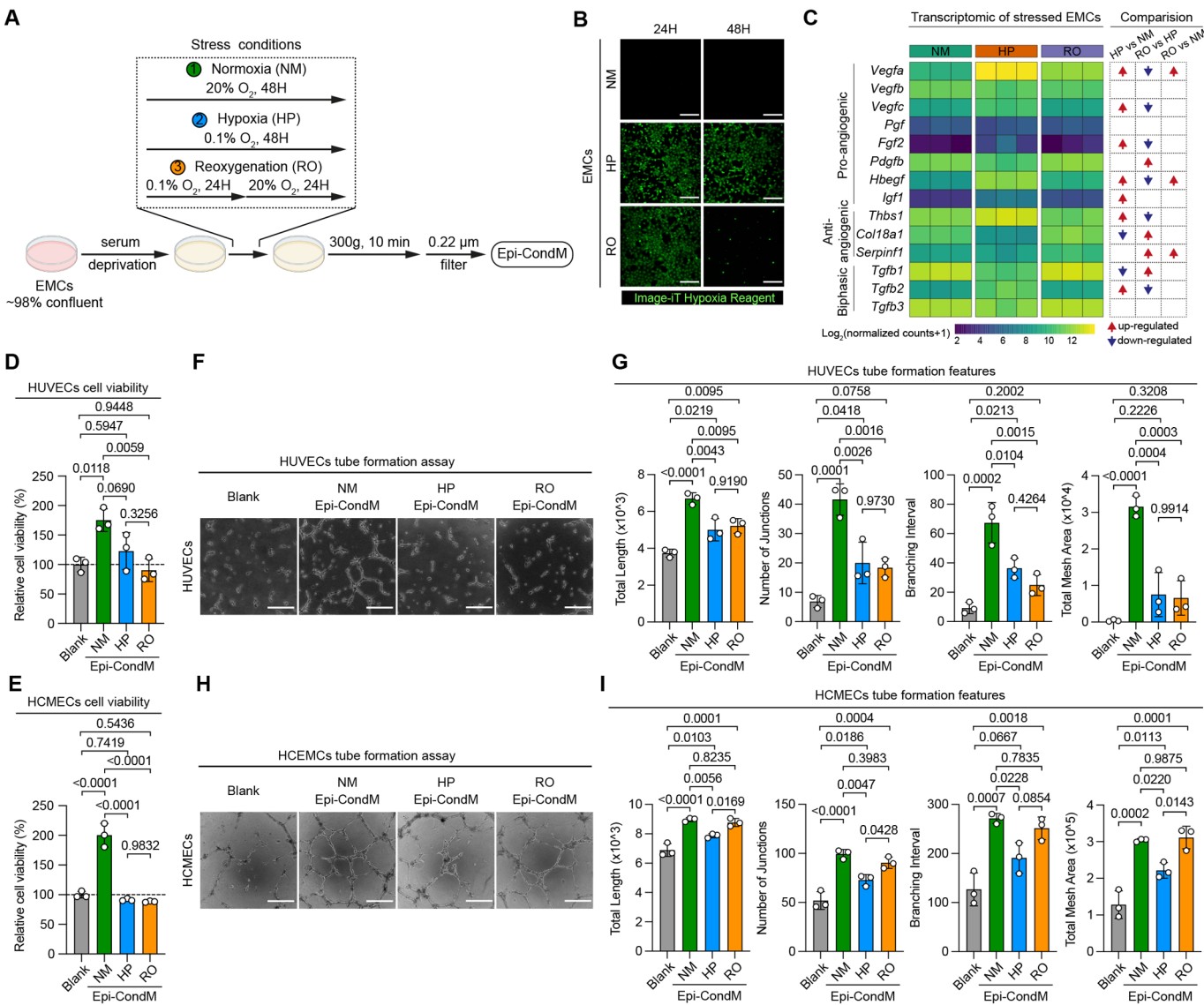

**Fig. 1. Hypoxic stress impaired the pro-angiogenic activities of Epi-CondM.** (A) Schematic representation of the experimental protocol for preparing hypoxic stress-induced Epi-CondM. Three oxygen conditions were used: NM, HP, and RO. (B) The hypoxic stress dynamics on rat EMCs were monitored using a HP detection reagent at 24 h and 48 h during the preparation process. Scale bars: 200 μm. (C) Transcriptomic analysis of angiogenic factors in EMCs. The left heatmap depicts the expression levels of pro-angiogenic, anti-angiogenic, and biphasic factors in EMCs under three conditions: NM, HP, and RO. Colour intensity represents the Log$_2$ (normalized counts+1) of gene expression. The right panel illustrates the differential expression comparisons between groups. Arrows indicate the direction of change (up-regulated, red; down-regulated, blue) for three pairwise comparisons: HP versus NM, RO versus HP, and RO versus NM. The raw data were downloaded from the ArrayExpress dataset E-MTAB-10594. (D,E) Evaluation of endothelial cell viability following incubation with stressed Epi-CondM using CCK-8 assay. Cellular viability was quantitatively assessed in HUVECs (shown in D) and HCMECs (shown in E) at 24 h post-treatment with stressed Epi-CondM. (F-I) Assessment of pro-angiogenic potential of stressed Epi-CondM using *in vitro* tube formation assay. Endothelial network formation was observed following 6 h incubation on the basement membrane extract. Representative images demonstrate tube formation in HUVECs (F) and HCMECs (H). Scale bars: 250 μm. Quantitative analysis of angiogenic parameters for HUVECs (G) and HCMECs (I) is presented.

both the sustained HP condition and initial HP phase of the RO condition exhibited a significantly induced hypoxic signal compared to the NM (Fig. S2A). Importantly, the subsequent reoxygenation phase in the RO group led to a significant reduction in the hypoxic signal at 48 h (Fig. S2A, $P$=0.0194 for RO-48H versus RO-24H). Collectively, these data confirm the successful induction of an intracellular hypoxic state in EMCs by the 0.1% O$_2$ exposure and the partial reversal of this state following reoxygenation. Since HP is often associated with cell stress and cell death (Lenihan and Taylor, 2013), we quantitatively examined the viability of EMCs by using the CCK8 assay (Fig. S2B). Exposure to HP severely compromised EMC

viability (HP versus NM, $P$<0.0001), while reoxygenation restored it ($P$<0.0001, RO versus HP; $P$=0.7072, RO versus NM).

We further examined the transcriptomic profiles of angiogenic genes in EMCs under NM, HP, and RO conditions (Fig. 1C). As expected, compared to the NM EMCs, HP EMCs significantly u-regulated key pro-angiogenic factor gene expression, including *Vegfa*, *Vegfc*, *Fgf2*, and *Hbegf*. Also as expected, the upregulated gene expression was partially reverted during RO.

We then collected conditioned media (Epi-CondM; Fig. 1A) from these epicardial cultures, determined the level of total protein in the samples (Fig. S3), and assayed the effects of normalized amounts of

Epi-CondM on two primary endothelial cell cultures, human umbilical vein endothelial cells (HUVECs) and human cardiac microvascular endothelial cells (HCMECs). As shown by CCK8 assay, NM Epi-CondM significantly enhanced endothelial cell viability and proliferation relative to blank media controls ($P<0.05$) in both HUVECs and HCMECs (Fig. 1D,E). In contrast, HP and RO Epi-CondM failed to elicit such an induction of proliferation (Fig. 1D,E). Critically, the viability elicited by both HP Epi-CondM and RO Epi-CondM was statistically indistinguishable from the blank media control in both HUVECs and HCMECs (all $P>0.05$). Furthermore, no significant difference was observed between the HP and RO Epi-CondM themselves ($P>0.05$), demonstrating that the loss of pro-proliferative activity persists after reoxygenation.

Tube formation assays demonstrated that NM Epi-CondM significantly enhanced endothelial network formation in both HUVECs and HCMECs compared to blank media controls, as evidenced by four key parameters: total length ($P<0.0001$), number of junctions ($P<0.001$), branching interval ($P<0.001$), and total mesh area ($P<0.001$) (Fig. 1F-I). In stark contrast, HP Epi-CondM did not display the angiogenic effect observed with NM Epi-CondM on these same parameters ($P<0.01$ versus NM for total length and number of junctions; $P<0.05$ versus NM for branching interval and total mesh area). The effect of RO Epi-CondM was endothelial-cell-type dependent. Thus, while RO Epi-CondM partially restored network-induction in HCMECs ($P<0.05$ for all parameters except for branching interval), the restorative effect was absent

in HUVECs ($P>0.05$ for all parameters). Thus, unexpectedly, the induction of angiogenic factors upon HP did not elicit a comparable increase in *in vitro* tube formation activity during the time observed.

## HP induces glycan remodelling of the epicardial secretome

The preceding experiments paradoxically showed that severe HP decreased angiogenic potential of epicardial secretome despite increased expression of angiogenic factors (Fig. 1). To explore a possible mechanism, we considered whether post-translational protein modification might be responsible. Since glycosylation figures prominently in the control of protein stability, secretion, and receptor-binding affinity (Chatham and Patel, 2024; Zhou and Qiu, 2019), we tested whether severe hypoxic stress might alter the glycosylation state of the epicardial secretome.

Taking advantage of the exquisite specificity and high affinity of lectins for particular carbohydrate moieties (Bojar et al., 2022), we profiled distinct glycan modifications of the epicardial secretome using a panel of 17 lectins (Fig. S4). Lectin-ELISA profiling revealed a hierarchical pattern of glycan interactions (Fig. 2A). Strong binding intensities in Epi-CondM were observed for lectins SNA, MAL-II, ConA, DBA, RCA, PSA, LCA, PHA-E, and Jacalin, whereas PHA-L, GS-IB4, s-WGA, SBA, and PNA exhibited weak binding. HP-induced alterations were consistently detectable within each lectin's dynamic range, demonstrating the robustness and sensitivity of our glycan profiling approach.

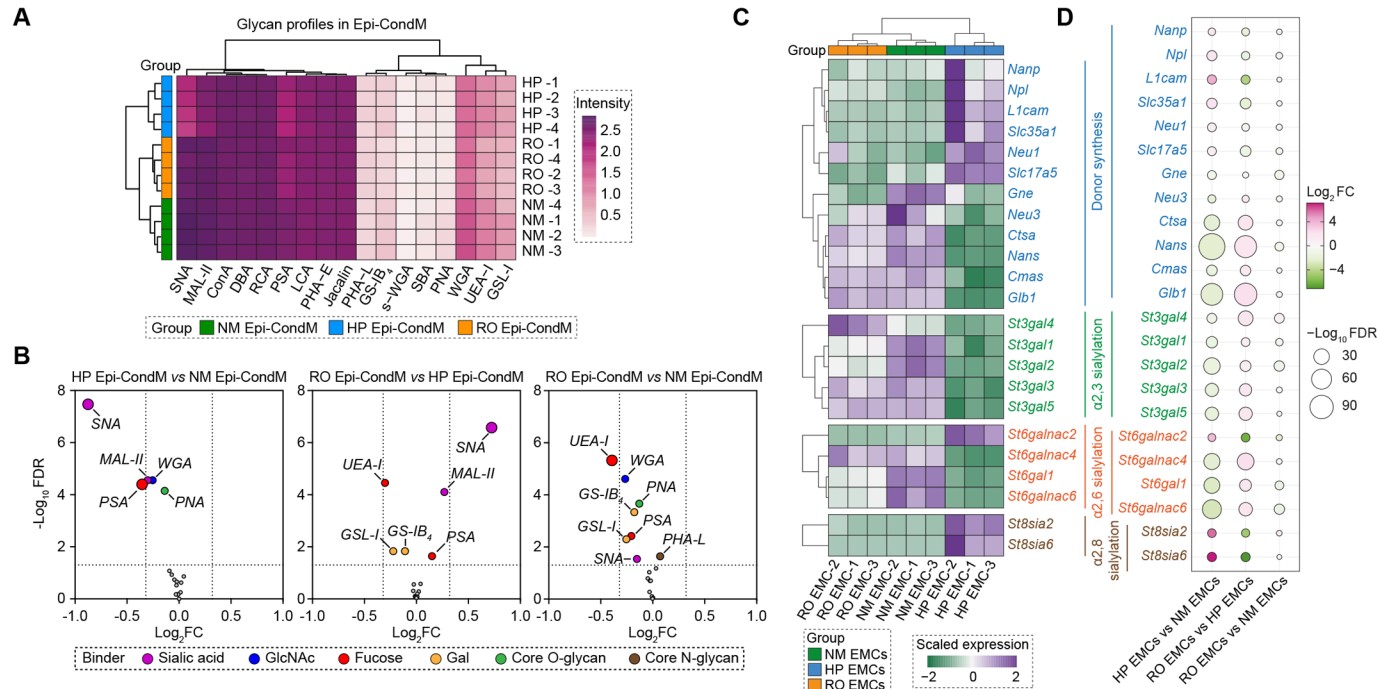

**Fig. 2. Characterization of glycan alterations in epicardial secretome.** (A,B) Lectin-based glycan profiling of the epicardial secretome. (A) Heatmap visualization of lectin-based enzyme-linked immunosorbent assay (Lectin-ELISA) profiling in stressed Epi-CondM. Binding intensities of 17 distinct lectins were measured at OD450 nm, with both lectins and samples hierarchically clustered. The colour gradient represents binding intensity, with magenta indicating high and white indicating low absorbance values. (B) Volcano plot analysis of differential lectin binding patterns between comparative conditions: HP versus NM, RO versus HP, RO versus NM. The horizontal dashed line indicates the significance threshold (FDR<0.05), and vertical dashed lines represent the fold change thresholds (|FoldChange|> 0.32). Significantly altered lectins (FDR<0.05) are highlighted, with lectin-specific colours representing their glycan-binding specificities. (C,D) Expression landscape of sialylation-related genes. (C) Hierarchical clustering heatmap displaying expression patterns of genes involved in sialylation. Gene symbols are color-coded based on their specific roles in sialylation processes: donor synthesis (blue), α2,3-linkage sialyltransferases (green), α2,6-linkage sialyltransferases (orange), and α2,8-linkage sialyltransferases (brown). The colour gradient represents scaled expression values, with purple indicating high and green indicating low expression. (D) Bubble plot illustrating comparative expression changes of sialylation-related genes across different conditions. Bubble size corresponds to the statistical significance (-Log10FDR), while colour intensity represents Log2FC. The raw data were downloaded from the ArrayExpress dataset E-MTAB-10594.

Severe HP caused significant changes in glycan alterations (Fig. 2B). Relative to NM, HP reduced both $\alpha$2,6- and $\alpha$2,3-linked sialylation, apparent by decreased binding of SNA and MAL-II (Log$_2$FC<−0.32, FDR<0.05), accompanied by diminished fucosylation signals detected by PSA (Log$_2$FC<−0.32, FDR<0.05). We also observed a partial restoration of SNA binding following RO (RO versus HP, Log$_2$FC> 0.32, FDR<0.05), indicating increased $\alpha$2,6-sialylation, which might explain that RO Epi-CondM partially recovered the angiogenic activity compared to HP Epi-CondM, especially in HCMECs (Fig. 1). Persistent alterations in fucose presentation were also apparent in the RO versus NM secretomes, as shown by reduced UEA-I binding (Log$_2$FC<−0.32, FDR<0.05). The concordance between restoration of angiogenic potency and $\alpha$2,6-sialylation suggest that sialylation status might govern the angiogenic abilities of the epicardial secretome.

Examining mRNA abundance of the genes regulating donor synthesis (*Gne*, *Nans*, *Npl*, etc.) and specific sialyltransferases (*St3gal1-6*, *St6gal1-2*, *St6galnac1-6*, *St8sia1-6*) provided further evidence that glycosylation status of the epicardial secretome is regulated by HP (Fig. 2C,D). All of these enzymes, with the exception of *St6galnac2* and $\alpha$2,8 sialyltransferase genes, were downregulated by exposure to HP (HP versus NM) and were significantly upregulated upon RO (RO versus HP), establishing a clear pattern of reversible hypoxic suppression. In contrast, the $\alpha$2,8 sialyltransferase genes (*St8sia2* and *St8sia6*) were induced by hypoxic conditions and inhibited during RO. These findings suggest that oxygen levels control the enzymatic machinery responsible for $\alpha$2,3- and $\alpha$2,6-sialylation in the epicardial secretome.

### Sialylation of the epicardial secretome enables angiogenic potential

To establish a functional relationship between the expression of sialyltransferases and angiogenesis, we blocked sialylation with 3Fax-Peracetyl Neu5Ac (P-3Fax), a non-selective sialyltransferase inhibitor (Macauley et al., 2014; Rillahan et al., 2012). EMCs were initially pre-treated with either 100 μM P-3Fax or vehicle control (0.11% DMSO) for 72 h. After several washes and media replacement, EMCs were incubated for an additional 48 h under normoxic conditions to collect the Epi-CondM (Fig. 3A). P-3Fax treatment decreased both $\alpha$2,6-linked (SNA binding; 20.7% reduction, *P*<0.0001) and $\alpha$2,3-linked (MAL-II binding; 48.6% reduction, *P*<0.0001) sialylation in the EMCs, which was assessed by lectin-based flow cytometry (Fig. 3B). Lectin-ELISA further verified that P-3Fax decreased sialylation of secreted proteins in Epi-CondM (Fig. 3C,D). Moreover, preventing sialic acid addition exposed underlying glycan structures, leading to increased detection of core-O glycans (PNA binding), galactose (Gal, GSL-I and GS-IB$_4$ binding), and N-acetylgalactosamine (GalNAc, DBA binding) (Fig. 3D), confirming successful generation of hypo-sialylated Epi-CondM (P-3Fax Epi-CondM).

As compared to vector (DMSO Epi-CondM), P-3Fax Epi-CondM abolished the protective effect of Epi-CondM (Fig. 3E,F), reducing viability by 15.5% in HUVECs (*P*<0.0001 versus DMSO) and 48.2% in HCMECs (*P*<0.01 versus DMSO).

Remarkably, P-3Fax Epi-CondM completely abrogated these pro-angiogenic effects in HUVECs, reducing all morphometric parameters to baseline levels (*P*<0.0001 versus DMSO) (Fig. 3G,H). In HCMECs, P-3Fax Epi-CondM exhibited intermediate activity, partially preserving network formation capacity, while significantly impairing total length, junction formation and total mesh area compared to DMSO Epi-CondM (*P*<0.05) (Fig. 3I,J). These cell type-specific responses suggest differential sialylation dependence in macrovascular versus microvascular endothelial cells.

### *St6gal1*-mediated $\alpha$2,6-sialylation regulates the angiogenic potential of epicardial secretome

While P-3Fax treatment effectively reduced sialylation in epicardial cells, we cannot formally exclude partial carryover of the inhibitor into conditioned media. This issue was therefore addressed by a genetic approach to test whether specific sialylation linkages promote angiogenic activity of the epicardial cell secretome. Specific sialyltransferases attach sialic acid via $\alpha$2,3, $\alpha$2,6, or $\alpha$2,8 linkages to galactose or GalNAc on N- and O-glycan chains (Li and Chen, 2012). Of the sialyl linkages ($\alpha$2,3, $\alpha$2,6, or $\alpha$2,8), $\alpha$2,6 linkage is particularly important because it mediates proper folding, stability, and secretion of broad-spectrum glycoproteins (Garnham et al., 2019; Zhu et al., 2024), including the angiogenic factors identified in Fig. 1C. *St6gal1* is the primary enzyme responsible for generating $\alpha$2,6 linkages and was downregulated by HP and upregulated upon RO (Fig. 2D,E). siRNA-mediated knockdown of *St6gal1* in EMCs under normoxia (Fig. 4A,B) selectively impaired $\alpha$2,6-sialylation [65±8% reduction in $\alpha$2,6-sialylation (SNA binding; 49.6% reduction, *P*<0.0001] while maintaining normal $\alpha$2,3-sialylation levels (MAL-II binding; *P*>0.05) (Fig. 4C). This specificity was further confirmed by lectin-ELISA of Epi-CondM (Fig. 4D,E), which showed a significant decrease in $\alpha$2,6-sialylated proteins (Log$_2$FC=−0.38, FDR<0.05) without altering $\alpha$2,3-sialylation patterns. *St6gal1* knockdown Epi-CondM markedly reduced endothelial cell viability activity by 15.5% in HUVECs (*P*<0.0001 versus si-NC) and 48.9% in HCMECs (*P*<0.01 versus si-NC) (Fig. 4F,G), with viability of inert control siRNA treated Epi-CondM being similar to blank media control.

*St6gal1* knockdown also completely abolished the ability of epicardial conditioned media to promote endothelial tube formation by HUVECs, reducing all morphometric parameters to baseline levels (*P*<0.0001 versus si-NC), and severely impairing pro-angiogenic effects on HCMECs (*P*<0.01 versus si-NC) at 24 h (Fig. 4H-K). In contrast, inert control siRNA did not impair the ability of Epi-CondM to stimulate HUVEC and HCMEC to promote tube formation (total length, junction numbers, branching interval, and total mesh area were all improved relative to hypoxia (*P*<0.01 for all parameters; Fig. 4H-K).

Although additional sialyltransferases were transcriptionally regulated under severe hypoxic conditions, selective knockdown of *St6gal1* uniquely reduced $\alpha$2,6-linked sialylation while preserving $\alpha$2,3-linked glycans, indicating a non-redundant role for ST6GAL1 in conditioning the epicardial angiogenic secretome.

### DISCUSSION
Our study identifies severe hypoxic stress-induced remodelling of epicardial sialylation, specifically the loss of ST6GAL1-mediated $\alpha$2,6-linked sialic acid, as a previously unappreciated mechanism by which the epicardial secretome regulates cardiac angiogenesis. While glycosylation is increasingly recognized as a central regulator of vascular biology – modulating receptor activation, extracellular matrix organization, endothelial survival, and vessel stability – most prior work has focused on endothelial or tumour cell–intrinsic glycosylation (Chandler et al., 2019; Chatham and Patel, 2024; Loaeza-Reyes et al., 2021). In contrast, our findings highlight glycosylation of epicardial secreted factors as a determinant of the endothelial angiogenic response under ischemic stress.

A substantial body of evidence demonstrates that glycosylation regulates angiogenic signalling at multiple levels (Chandler et al., 2019; Imamaki et al., 2018; Zhong et al., 2020). N-glycans on vascular endothelial growth factor receptor 2 (VEGFR2), integrins, and other endothelial receptors modulate ligand binding, receptor

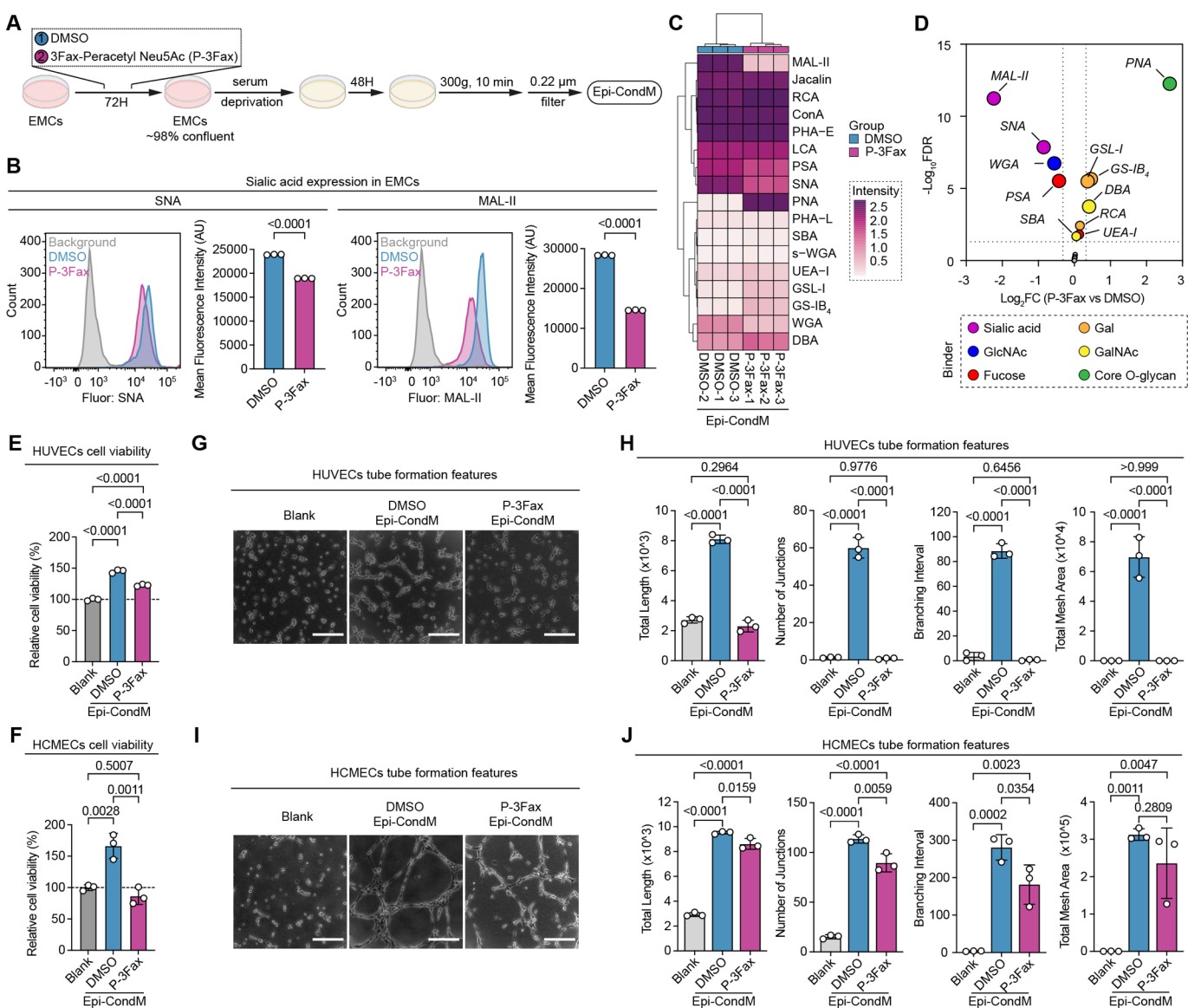

**Fig. 3. Sialyltransferase inhibition impaired the pro-angiogenic activities of Epi-CondM.** (A) Schematic illustration of the experimental workflow for generating sialic acid-depleted Epi-CondM. Sialylatransferase activity in EMCs was pharmacologically inhibited using the small molecule inhibitor P-3Fax, resulting in decreased secretion of sialic acid-modified proteins in Epi-CondM. (B) Verification of sialyltransferase inhibition in EMCs was performed using lectin-based flow cytometry analysis. The expression of sialic acid residues was assessed using two specific lectin probes: *Sambucus nigra* lectin (SNA) for α2,6-linked sialic acid detection and *Maackia amurensis* lectin II (MAL-II) for α2,3-linked sialic acid identification. Representative flow cytometry histograms and quantitative analysis of mean fluorescence intensity (MFI) are presented. (C,D) Verification of reduced secretion of sialic acid-modified proteins in Epi-CondM was conducted through lectin-based enzyme-linked immunosorbent assay (ELISA). The glycosylation profiles in Epi-CondM were visualized using a heatmap (C), while differential lectin binding patterns were illustrated in a volcano plot (D). (E,F) Evaluation of endothelial cell viability following incubation with Epi-CondM using CCK-8 assay. Cellular viability was quantitatively assessed in HUVECs (E) and HCMECs (F) at 24 h post-treatment with Epi-CondM. (G-J) Assessment of pro-angiogenic potential of Epi-CondM using *in vitro* tube formation assay. Endothelial network formation was observed following 6 h incubation on the basement membrane extract. Representative images demonstrate tube formation in HUVECs (G) and HCMECs (I). Scale bars: 250 μm. Quantitative analysis of angiogenic parameters for HUVECs (H) and HCMECs (J) is presented.

clustering, and downstream signalling, and targeted editing of these glycans alters VEGF-induced phosphorylation and endothelial sprouting (Chandler et al., 2019; Imamaki et al., 2018). Beyond receptor-level effects, global changes in endothelial glycosylation influence Notch, Wnt, and integrin pathways, thereby shaping morphogenesis and vessel maturation (He et al., 2024). Our data extend this framework upstream: rather than modifying endothelial receptors directly, hypoxic stress rewires the glycosylation of epicardial secreted proteins, altering how endothelial cells perceive and integrate pro-angiogenic cues.

Under NM conditions, the epicardial secretome is enriched in α2,6- and α2,3-sialylated glycans and robustly supports endothelial viability and tube formation. Despite transcriptional induction of classical angiogenic factors under HP, extreme hypoxic stress produces a paradoxical loss of angiogenic function (Fig. 1). Our data indicate that this disconnect arises from post-translational glycan remodelling (Fig. 2A,B), with a disproportionate loss of α2,6-linked sialylation. This observation is consistent with the concept that glycosylation functions as a 'second code' for angiogenesis, whereby qualitative changes in glycan structure can

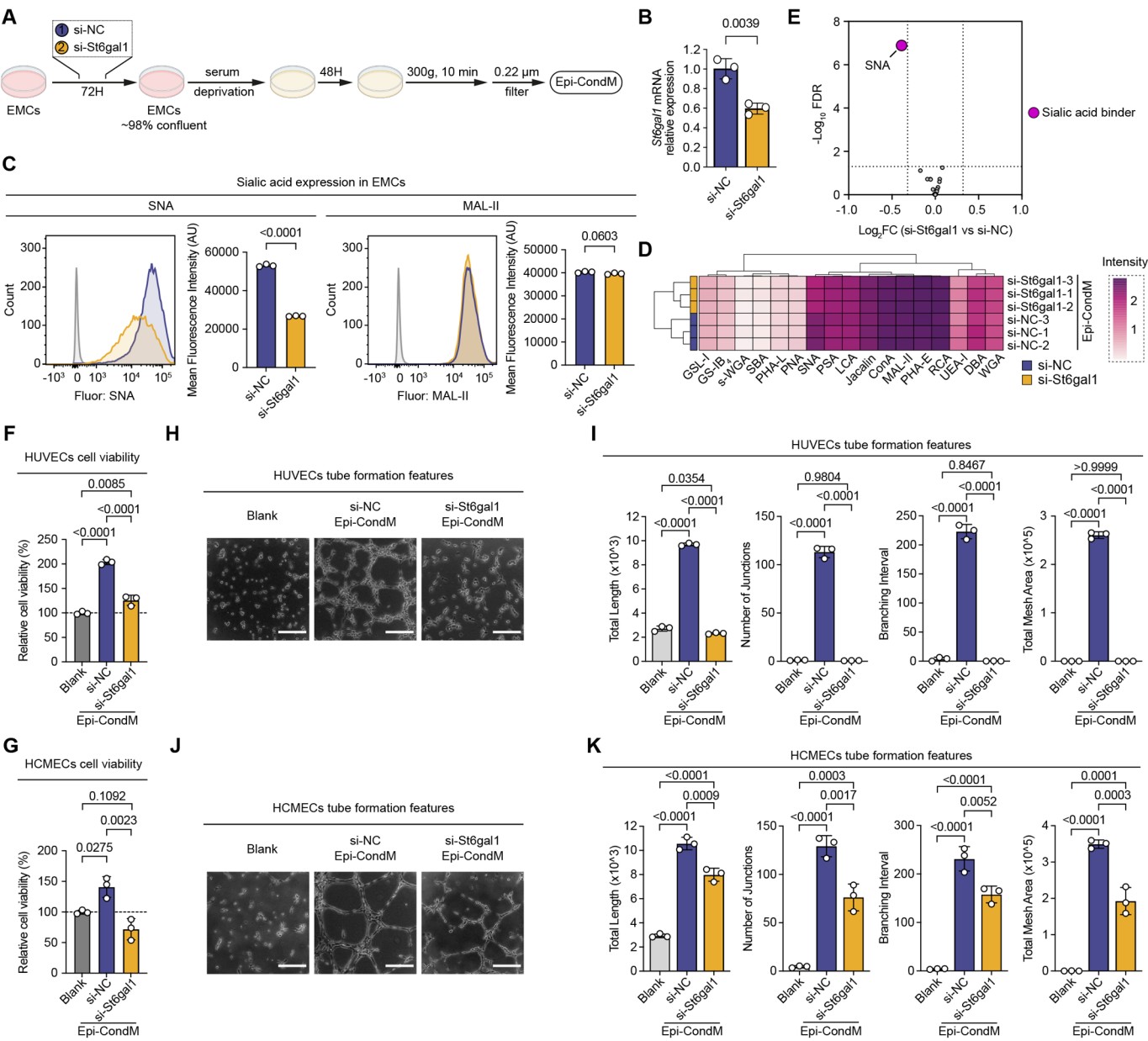

**Fig. 4. *St6gal1* knockdown disrupted the pro-angiogenic functions of Epi-CondM.** (A) Schematic illustration of the experimental workflow for generating *St6gal1* knockdown Epi-CondM. *St6gal1* activity in EMCs was genetically reduced using the siRNA for *St6gal1*, resulting in decreased secretion of sialic acid (especially for α2,6) -modified proteins in Epi-CondM. (B) Validation of St6gal1 knockdown in EMCs. qPCR analysis of St6gal1 mRNA expression following siRNA-mediated knockdown in EMCs. Data are normalized to Gapdh and presented as fold change relative to scramble control (si-NC). Data are shown in mean±SD, *n*=3; Student's *t*-test. (C) Verification of α2,6 sialic acid residues reduction in EMCs was performed using lectin-based flow cytometry analysis. The expression of sialic acid residues was assessed using two specific lectin probes: SNA and MAL-II. Representative flow cytometry histograms and quantitative analysis of MFI are presented. (D,E) Verification of reduced secretion of α2,6 sialic acid-modified proteins in Epi-CondM was conducted through lectin-based ELISA. The glycosylation profiles in Epi-CondM were visualized using a heatmap (D), while differential lectin binding patterns were illustrated in a volcano plot (E). (F,G) Evaluation of endothelial cell viability following incubation with Epi-CondM using CCK-8 assay. Cellular viability was quantitatively assessed in HUVECs (F) and HCMECs (G) at 24 h post-treatment with Epi-CondM. (H,K) Assessment of pro-angiogenic potential of Epi-CondM using *in vitro* tube formation assay. Endothelial network formation was observed following 6 h incubation on the basement membrane extract. Representative images demonstrate tube formation in HUVECs (H) and HCMECs (J). Scale bars: 250 µm. Quantitative analysis of angiogenic parameters for HUVECs (I) and HCMECs (K) is presented.

shift biological outcomes even when the underlying protein repertoire remains intact.

Within the glycome, ST6GAL1-mediated α2,6-sialylation has been repeatedly implicated in vascular regulation (Garnham et al., 2019). In tumour models, loss of ST6GAL1 in endothelial cells disrupts PECAM-VEGFR2-integrin complexes, enhances apoptosis, and suppresses angiogenesis (Imamaki et al., 2018), whereas

increased ST6GAL1 activity promotes endothelial survival and aberrant neovascularization (Gc et al., 2022). Our findings are consistent with a pro-angiogenic role for ST6GAL1 but reveal a distinct cellular compartment in which it operates. Rather than acting within endothelial cells, ST6GAL1 in epicardial mesothelial cells conditions the glycosylation – and thus the bioactivity – of paracrine angiogenic signals. HP suppresses *St6gal1* expression in EMCs

and α2,6-sialylation in the epicardial secretome (Fig. 2), and selective *St6gal1* knockdown under NM conditions phenocopies this loss (Fig. 4), abolishing the capacity of epicardial conditioned media to support endothelial survival and network formation.

The relationship between HP, glycosylation, and angiogenesis is highly context dependent (Croci et al., 2014b; Hashimoto and Shibasaki, 2015; Shirato et al., 2010). HP is classically pro-angiogenic through HIF-dependent induction of VEGF and related factors (Hashimoto and Shibasaki, 2015); however, it also alters nucleotide sugar metabolism and glycosyltransferase expression (Shirato et al., 2010). In tumours, HP-driven glycan remodelling can preserve angiogenesis via alternative lectin-mediated pathways (Croci et al., 2014a). In contrast, our data show that severe hypoxic stress suppresses ST6GAL1 expression in epicardial cells, thereby blunting paracrine angiogenic support in the ischemic heart. These findings emphasize that glycan signatures promoting angiogenesis in cancer may not be equivalent to those required for reparative angiogenesis following myocardial infarction.

From a translational perspective, these findings position epicardial glycosylation as a modifiable determinant of post-ischemic angiogenesis. While global inhibition of glycosylation has been explored as an anti-angiogenic strategy in oncology (Garnham et al., 2019), our selective perturbation of ST6GAL1 in epicardial cells avoids the widespread protein misfolding and ER stress associated with pan-glycosylation blockade, instead revealing a fine-tuning role for α2,6-sialylation in shaping the reparative microenvironment.

## Limitations

This study has several limitations. First, although lectin-based and functional assays demonstrate robust remodelling of secreted glycans, we did not perform glyco-enriched proteomics to identify individual sialylated factors responsible for the observed effects. Second, gain-of-function or rescue experiments restoring ST6GAL1 activity under hypoxic stress would further strengthen causal inference. Third, all findings were generated *in vitro*, and *in vivo* validation in ischemic heart models will be necessary to establish translational relevance. Despite these limitations, our data establish ST6GAL1-mediated α2,6-sialylation as a necessary determinant of epicardial paracrine angiogenic support under extreme hypoxic stress.

## MATERIALS AND METHODS
### Cell culture
#### EMCs
Primary rat EMCs were isolated from the epicardium layer of Sprague-Dawley (SD) rat hearts (Eid et al., 1992). EMCs were maintained in low-glucose DMEM (5.5 mM D-glucose) with 10% fetal bovine serum (FBS), 1% L-Glutamine, and 1% penicillin/streptomycin (10,000 U/ml) at 37°C with 5% $CO_2$. These cells were characterized as EMCs by highly expressed WT1 and the other epicardial markers (Fig. S1 and Table S1). EMCs from passages 25 to 30 were used in this study.

For all endothelial functional assays, epicardial conditioned media (Epi-CondM) were applied undiluted and normalized by total protein concentration as determined by BCA assay (Fig. S3). Equal protein input was confirmed across all experimental conditions prior to endothelial cell treatment.

#### HUVECs
HUVECs isolated from two individual donors (Lot# 2422001 and Lot# 2394291, Thermo Fisher) were maintained in Human Large Vessel Endothelial Cell Basal Medium (Gibco) supplemented with Large Vessel Endothelial Supplement (LVES, Gibco) according to the manufacturer's instructions. The HUVECs were cultured at 37°C with 5% $CO_2$ and fed every 2-3 days. HUVECs from passage 3 to 6 were used in this study.

#### HCMECs
HCMECs isolated from two individual healthy donors (Lot# 492Z009.4 and Lot# 488Z023.3, PromoCell) were maintained in Endothelial Cell Growth Medium MV (PromoCell) according to the manufacturer's instructions. The HCMECs were cultured at 37°C with 5% $CO_2$ and feed every 2-3 days. HCMECs from passage 4 to 6 were used in this study.

### Epi-CondM preparation
#### Stressed Epi-CondM preparation
For stressed Epi-CondM preparation, rat EMCs were expanded in 0.1% gelatine-coated 10 cm Petri dishes until reaching ∼98% confluency, then softly washed with warm PBS twice to remove remaining serum (FBS) components. Media was subsequently changed to 10 ml serum-free media (low-glucose DMEM with 1% L-Glutamine, and 1% penicillin/streptomycin) and then these EMCs were cultured under in the hypoxic workstation (Whitley H35 Hypoxystation, Don Whitley Scientific) under different oxygen concentrations for 48 h before conditioned media collection. NM group was cultured under 20% $O_2$ for 48 h, and HP group was cultured under 0.1% $O_2$ for 48 h. In addition, the RO group was cultured under 0.1% $O_2$ for 24 h, subsequently cultured under 20% $O_2$ for 24 h, to simulate ischemia-reperfusion. Crude conditioned media were centrifuged at 300 *g* for 10 min to remove dead cells, and supernatants were subsequently filtered through the PSE membrane filter (0.22 μm, Millipore) to remove cell debris. Finally, Epi-CondM was aliquoted and stored at −80°C until use. In parallel, three dishes with 10 ml serum-free media per dish were kept under 20% $O_2$ for 48 h as the blank control.

#### Hypo-sialylated Epi-CondM preparation
P-3Fax was used as the inhibitor to suppress sialyltransferase activity (Macauley et al., 2014), thereby reducing the production and secretion of sialic acid. P-3Fax (Sigma) was reconstituted in DMSO to prepare a stock solution (90 mM) following the manufacturer's instruction and subsequently diluted to a final concentration of 100 μM in EMCs maintenance medium (low-glucose DMEM with 10% FBS, 1% L-Glutamine, and 1% penicillin/streptomycin). EMCs were incubated for 3 days in the presence or absence of 100 μM P-3Fax-NeuAc (0.11% of DMSO as the vector control). Afterwards, the culture medium was replaced with serum-free medium, and Epi-CondM was collected over the next 48 h. The procedure for processing crude conditioned media was performed as described above.

#### *St6gal1* knockdown Epi-CondM preparation
To suppress St6gal1 expression in EMCs, a combination of two Silencer™ Select pre-designed siRNAs (s129173 and s129174, Invitrogen) targeting *St6gal1* was transfected into EMCs with Lipofectamine RNAiMAX Reagent (Invitrogen) according to the manufacturer's instructions. Cells were transfected with siRNAs in complete growth medium for 3 days to ensure effective knockdown. Two Silencer™ Select negative control siRNAs (Negative Control No.1 #4390843 and Negative Control No.2 #4390845, Invitrogen) were used as the negative control. Following this incubation period, the medium was replaced with serum-free medium, and crude conditioned media were collected over the subsequent 48 h. Crude conditioned media were processed as described above. After conditioned media collection, the remaining cells were harvested to validate *St6gal1* knockdown in EMCs (primers used are listed in Table S2).

### Assessment of hypoxic stress in EMCs
To assess hypoxic stress in EMCs, oxygen level was monitored using the Image-iT™ Green Hypoxia Reagent (Thermo Fisher) according to the manufacturer's instructions. At 0 h, 24 h, and 48 h during preparation of stressed Epi-CondM, a randomly selected culture dish from each experimental group (NM, HP, and RO) was subjected to HP detection. EMCs were incubated with Image-iT™ Green Hypoxia Reagent at a final concentration of 5 μM in pre-warmed serum-free media for 30 min at 37°C in the dark. After incubation, EMCs were washed twice with PBS to remove unbound reagent. For fluorescence detection, EMCs were imaged using an inverted microscope (LSM-780, Zeiss) equipped with a FITC filter set, and images were captured at ×10 magnification.

## Quantitative PCR (qPCR)

Total RNA was extracted from cultured rat EMCs using QIAzol Lysis Reagent (Qiagen) and RNeasy Mini Kit (Qiagen). RNA concentration and purity were determined by a Nanodrop 2000 spectrophotometer (Thermo Fisher). For each sample, 1 µg of total RNA was reverse-transcribed into cDNA using PrimeScript RT Master Mix (Takara Bio) following the manufacturer's protocol. qPCR was performed on a QuantStudio 6 Flex Real-Time PCR System (Applied Biosystems) using TB Green Premix Ex Taq (Takara Bio). This workflow was applied to: (1) verify the epicardial identity of cultured EMCs by examining marker genes, and (2) evaluate the knockdown efficiency of *St6gal1* in EMCs. Relative gene expression was calculated using the ΔΔCT method (Schmittgen and Livak, 2008) and normalized to *Gapdh* as an internal control. Non-epicardium controls (LV and RV myocardium) were kindly provided by Prof. Cesare Terracciano (Imperial College London, UK). All primer sequences are summarized in Tables S1 and S2.

## Cell viability assay with CCK8

Cell viabilities of EMCs, HUVECs, and HCMECs were determined by Cell Counting Kit 8 (CCK8, Abcam). Cells were seeded into the 96-wells plate at a density of $2 \times 10^5$ cells/cm$^2$ and maintained until reached 60% confluency. The cells were then treated with 100 µl Epi-CondM for 24 h. After adding 10 µl CCK8 solution into the cell culture medium, the plate was incubated at 37°C for 2 h, and then the absorbance at 460 nm was measured using a CLARIOstar Plus plate reader (BMG LABTECH). Absorbance at 460 nm is highly correlated with live cell number.

## Tube formation assay

HUVECs and HCMECs were harvested and resuspended at a density of $2 \times 10^5$ cells/ml in endothelial growth medium. A 10 µl aliquot of Geltrex™ LDEV-Free Reduced Growth Factor Basement Membrane Matrix (Gibco) was used to coat each well of a µ-Slide 15 Well 3D slide (Ibidi) according to the manufacturer's instructions. 5 µL aliquot of cell suspension and 50 µL Epi-CondM were then added onto the polymerized Geltrex™ matrix and incubated at 37°C under 5% CO$_2$. After 6 h of incubation, endothelial network was imaged using a Zeiss Axio Observer widefield microscope at 5× magnification. Quantitative analysis of angiogenesis was performed using the Angiogenesis Analyzer plugin (Carpentier et al., 2020) for NIH FIJI software. Key parameters, including total length, number of junctions, branching interval, and total mesh area were quantified to evaluate angiogenic potential.

## Glycan profiling by lectin-based enzyme-linked immunosorbent assay (ELISA)

Glycan profiles in Epi-CondM were analysed using a lectin-based ELISA assay. Briefly, Epi-CondM samples were diluted 1:10 in 0.1 M bicarbonate/carbonate coating buffer (pH 9.6), and 50 µl of diluted samples were immobilized per well in a 96-well Maxisorp™ microtiter plate by overnight incubation at 4°C. Plates were blocked with 200 µl of 1× Carbo-Free™ blocking buffer (SP-5040-125, Vector Laboratories) for 2 h at room temperature to minimize non-specific binding. After five washes with 200 µl of TBS-T (Tris-buffered saline with 0.05% Tween-20), 50 µl of biotinylated lectin solutions (reconstituted in 2 µg/ml with TBS-T, lectins purchased from Vector Laboratories) was added to each well and incubated for 30 min at room temperature. Unbound lectins were removed by four additional TBS-T washes, followed by incubation with 50 µl of HRP-conjugated streptavidin (1:10,000 dilution in TBS-T, Proteintech) for 30 min. Plates were washed five times with TBS-T and following once with PBS, and subsequently 100 µl of TMB substrate (ab171523, pre-equilibrated to room temperature) was added to each well. The reaction was allowed to proceed for 15 min and stopped by adding 100 µl of TMB stop solution (ab171529, Abcam). Absorbance was measured at 450 nm and 560 nm using a two-wavelength read method to correct for background interference. A panel of 17 lectins was used to comprehensively profile major glycan motifs (Fig. S4).

## Determination of cellular sialic acid expression by lectin-based flow cytometry

Lectin-based flow cytometry was used to determine the sialic acid expression on EMCs. In briefly, EMCs were harvested and blocked with Carbo-Free™ blocking solution for 30 min at on ice to minimize non-specific interactions. Cells were then stained with 10 µg/ml biotinylated lectins (SNA or MAL-II) diluted in TBS-T for 30 min on ice. Unbound lectins were removed by washing cells twice with PBS, followed by incubation with Alexa Fluor 488-conjugated streptavidin (Thermo Fisher) for 15 min in the dark. Cells were washed again with PBS and resuspended in PBS for analysis. Data acquisition was performed using a FACSymphony™ A3 Cell Analyzer (BD Biosciences), and analysis was conducted using FlowJo software (version 10.9.0, BD Life Sciences). Unstained cells were included as background controls while cells only stained with Alexa Fluor 488-conjugated streptavidin were used to adjust the voltage. The mean fluorescence intensity (MFI) was used to evaluate the sialic acid expression in the EMCs.

## Bioinformatic analysis of transcriptomic data

The transcriptomic data of EMCs under NM, HP, and RO, were analysed using raw data retrieved from the ArrayExpress database (accession number: E-MTAB-10594). Following normalization, differential expression analysis was performed to identify transcriptional shifts across conditions by DESeq2 (Love et al., 2014). Specific attention was directed toward angiogenic factors and the enzymatic machinery governing sialylation. Visualization of these expression patterns was implemented through hierarchical clustering, volcano plots, and bubble plots to characterize the glycan-related transcriptional landscape.

## Statistics analysis

Statistical analyses were performed using GraphPad Prism 9 (GraphPad Software, Inc.). For comparisons across multiple groups, one-way ANOVA with appropriate post-hoc tests was applied, with statistical significance set at $P < 0.05$ unless otherwise specified. All quantitative data are presented as mean±s.d. (s.d.) unless otherwise indicated in the figure legends.

### Acknowledgements

The authors thank the Imperial College London core facilities for their important contributions to this work, particularly the Facility for Imaging by Light Microscopy (FILM) for assistance with widefield microscopy. The authors also acknowledge the China Scholarship Council (CSC) and Imperial College London for providing the PhD studentship to J.Z.

### Competing interests

P.R.L. holds equity in Regencor, Inc. The authors have no other competing interests.

### Author contributions

Conceptualization: P.R.-L., J.Z., C.E.; Formal analysis: P.R.-L., J.Z., C.E.; Funding acquisition: P.R.-L., J.Z., C.E.; Investigation: P.R.-L., J.Z., H.J.W.; Methodology: J.Z., H.J.W.; Project administration: P.R.-L.; Supervision: P.R.-L.; Validation: J.Z.; Writing – original draft: J.Z.; Writing – review & editing: P.R.-L., C.E.

### Funding

This work was supported by the California Institute for Regenerative Medicine [CIRM TRAN-12097 to P.R.L.]; and British Heart Foundation [PG/22/11083 to P.R.L.]. Facility for Imaging by Light Microscopy (FILM) is part-supported by funding from the Wellcome Trust [104931/Z/14/Z] and the Biotechnology and Biological Sciences Research Council [BB/L015129/1]. Infrastructure support was provided by the National Institute for Health Research (NIHR) Imperial Biomedical Research Centre (BRC). H.J.W. would like to acknowledge MRC Equip – World Class Labs award MC_PC_MR/X013537/1. Open Access funding provided by Regencor. Deposited in PMC for immediate release.

### Data and resource availability

The bulk transcriptomic data used in this study were retrieved from the ArrayExpress database under accession number E-MTAB-10594. All relevant data and details of resources can be found within the article and its supplementary information.

### Peer review history

The peer review history is available online at https://journals.biologists.com/bio/lookup/doi/10.1242/bio.062479.reviewer-comments.pdf

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
