## [Peer Review File · Biology Open]

Severe hypoxia drives loss of ST6GAL1-mediated α 2,6-sialylation in the epicardial secretome impairing angiogenic activity

Junqing Zhang, Harry J. Whitwell, Costanza Emanuelli and Pilar Ruiz-Lozano
10.1242/bio.062479

Editor: Tristan Rodríguez

Review timeline

Original submission:	9 January 2026
Editorial decision:	19 January 2026
First revision received:	27 March 2026
Accepted:	30 March 2026

Original submission

First decision letter

MS ID#: bio.062479

MS Title: Severe hypoxia drives loss of ST6GAL1-mediated α 2,6-sialylation in the epicardial secretome impairing angiogenic activity

Authors: Pilar Ruiz-Lozano, Junqing Zhang, Harry J. Whitwell and Costanza Emanuelli

I have now reached a decision on the above manuscript.

The reviewer reports are shown at the bottom of this email.

As you will see, the reviewers raised a number of criticisms that prevent me from accepting the paper at this stage. They suggest, however, that a revised version might prove acceptable, if you can address their concerns. I believe that the vast majority of the reviewers' comments can be addressed by modifying the claims presented in the manuscript. These should be adjusted so that they are more circumspect, and more accurately reflect the results that are presented in the manuscript. If you think that you can deal satisfactorily with the criticisms on revision, I would be pleased to see a revised manuscript.

At this stage, we also ask you to ensure your manuscript complies with our formatting guidelines. Provided you are able to fully address the referees' comments, we are positive about publication of your paper (we accept over 95% of revision submissions) and therefore hope you won't mind any extra work involved in reformatting your manuscript at this point.

Please upload both a 'clean' version of your Word file, along with a highlighted version clearly showing where you have made changes in the revised manuscript. Please avoid using 'Track changes' in Word files as these are lost in PDF conversion.

I should be grateful if you would also provide a point-by-point response detailing how you have dealt with the points raised by the reviewers in the 'Response to Reviewers' box. Please attend to all of the reviewers' comments. If you do not agree with any of their criticisms or suggestions please explain clearly why this is so.

Reviewer 1

Comments for the author

The manuscript by Zhang et al. addresses an interesting and timely question in cardiac repair, namely how severe hypoxia alters the epicardial secretome and its ability to support endothelial survival and angiogenesis. The study is well executed and the data support the central observation that epicardial conditioned media generated under 0.1% O₂ has reduced pro-angiogenic activity, alongside changes in sialylation. However, a few clarifications and some softening of mechanistic and translational claims are needed to ensure the conclusions are fully supported by the data presented and the work is readily reproducible. Please see my specific comments below:

it would be of value to strengthen the reporting of biological replicates and donor handling, which is key for reproducibility. The endothelial assays use HUVECs and HCMECs from two donors each, which is a good start, but it is not always clear whether n refers to independent EMC isolations/conditioned media preparations, technical repeats, endothelial donors or wells/images. Please state explicitly for each main figure (i) the number of independent epi-condm preparations as biological replicates, (ii) whether analyses were performed per donor and then pooled or otherwise and (iii) how donor-to-donor variability was handled statistically.

-while the choice of epicardial mesothelial cells and both HUVECs and HCMECs is widely used in the field, the rationale for selecting these specific in vitro models is largely implicit. We would welcome a brief justification of their relevance and limitations to further strengthen the experimental framework.

-the model relies on very late passage "primary", where passages 25-30 were used. This is unusually high for primary cells and could materially affect secretory phenotype and stress responses. Please justify and/or control for this- at minimum, please justify why this passage range is appropriate and consider adding a sensitivity analysis, e.g. repeating a key experiment with earlier passages or independent isolations.

-the results state that the loss of pro-proliferative activity persists after reoxygenation (HP and RO indistinguishable in viability assays, Fig 1D-E), yet tube formation shows partial restoration in HCMECs with RO Epi-CondM (Fig 3G-J). The lectin section suggests restored α 2,6-sialylation "might explain" the recovered angiogenic activity. These are not contradictory, but the manuscript should more clearly separate (i) proliferation/viability effects and (ii) network formation effects. This would avoid implying a single unified "recovery" phenotype without specifying which functional readout.

-it is a particular strength of the manuscript that the authors include a clear and balanced "limitations" section, which appropriately contextualises the in vitro nature of the work and avoids overstatement of translational impact. Indeed, the suggested mechanistic causality is still incomplete as acknowledged by the authors in this section - while LOF evidence is supportive of this, causal inference would be stronger with a rescue set up. Therefore we would ask the authors to temper claims and/or add a rescue experiment on the conclusion that ST6GAL1-dependent α 2,6-sialylation is required revealing a new therapeutic target. We would recommend softening the abstract and discussion language to "suggests/consistent with/may represent" unless a rescue is added. Related to the this, the "therapeutic target" and in vivo relevance are currently speculative given all findings are in vitro. We recommend toning down statements implying translational readiness in the abstract and conclusions and framing therapeutic relevance as a hypothesis for future testing.

-please clarify the hypoxia monitoring replication and sampling strategy. Oxygen level monitoring uses a randomly selected culture dish from each group. This reads as n=1 dish per condition/timepoint, which is not ideal. Please report how many independent dishes/experiments were quantified for hypoxia reagent imaging and for EMC viability (Fig S2), and whether the "random dish" approach was repeated across independent runs.

-the transcriptomic re-analysis is based on ArrayExpress E-MTAB-15094, please provide sufficient parameters for re-analysis such as group definitions, threshold and any batch handling that was performed during the analysis.

Reviewer 2

Comments for the author

The authors' data supports some of the conclusions, but given the limitations of the study, there is insufficient evidence that this mechanism plays any role in the epicardial response upon ischemic injury.

This model relies entirely on in vitro data, therefore drawing conclusions about the actual role of epicardium signalling in ischemic (or IR) injury is very unlikely to be reflected. Moreover, the authors combine two different model organisms and do not address whether this response is maintained across species (and very different stages of development, as is the case between this epicardial cell line and HUVECs).

The epicardial cell line used presents specific caveats: it is a dormant epicardium isolated from adult mouse heart. Therefore, it has been immortalized in an epithelial state. It is known that the epicardium undergoes EMT and becomes activated, both in development and in response to injury and it is this specific phenotype that presents the most relevant paracrine signalling. Moreover, it is highly unlikely that these severe hypoxic levels would be reached (even in extreme ischemic injury) in the outermost layer of the heart (where epicardial cells in epithelial state reside).

In order to address this major issue, the ideal scenario would be to address epicardial secretome in vivo after hypoxic conditions/ischemic damage and whether sialylation is also taking place in a significant manner (alternatively, Stgal1 levels could be evaluated).

Alternatively, addressing if this response is conserved, primary epicardial cells from mouse or human model could be cultured and their Epi-CM evaluated for PTM under severe hypoxic conditions.

Another point that should be addressed how the diminishing of cell viability upon extreme hypoxia of epicardial cells is behind the response and changes in CM. Some other stress-inducing affecting variability could also be tested.

If primary epicardial cultures are evaluated, how does extreme hypoxia alter specifically the activation phenotype of epicardial cells and does that impact its secretome?

Lastly, the authors evaluate transcription of proangiogenic factors (Such as VEGFa and c, and Fgf2) upon severe hypoxia of epicardial cells however the proangiogenic response is abrogated. However, the presence of these factors in the conditioned media is not evaluated.

Reviewer's Responses to Questions

Experimental quality

Does each figure have the proper controls?

If 'No', please indicate reasons in Comments for Author box below.

Reviewer #1:

- Yes

Reviewer #2:

- Yes

Were the data analyzed using appropriate statistical tests?

If 'No', please indicate reasons in Comments for Author box below.

Reviewer #1:

- Yes

Reviewer #2:

- Yes

Reproducibility

Were experiments performed using adequate number of biological replicates?

If 'No', please indicate reasons in Comments for Author box below.

Reviewer #1:

- No

Reviewer #2:

- Yes

Does the methods section provide sufficient detail to permit reproducibility?

If 'No', please indicate reasons in Comments for Author box below.

Reviewer #1:

- No

Reviewer #2:

- Yes

Completeness

Are the manuscript's conclusions supported by the data?

If 'No', please indicate reasons in Comments for Author box below.

Reviewer #1:

- Yes

Reviewer #2:

- No

Scholarship

Do the authors cite and discuss the merits of data that would argue for and against their conclusion?

If 'No', please indicate reasons in Comments for Author box below.

Reviewer #1:

- Yes

Reviewer #2:

- No

Does the manuscript title & abstract accurately reflect the contents of the manuscript, without hyperbole?

If 'No', please indicate reasons in Comments for Author box below.

Reviewer #1:

- No

Reviewer #2:

- Yes

First revision

Author response to reviewers' comments

We thank reviewers for their careful evaluation of our manuscript and for their constructive comments. We appreciate the recognition of the study's strengths and the opportunity to improve clarity, rigor, and framing. Below, we provide a detailed, point-by-point response to all comments.

Comments from the Reviewers:

Reviewer 1: The manuscript by Zhang et al. addresses an interesting and timely question in cardiac repair, namely how severe hypoxia alters the epicardial secretome and its ability to support endothelial survival and angiogenesis. The study is well executed and the data support the central observation that epicardial conditioned media generated under 0.1% O₂ has reduced pro-angiogenic activity, alongside changes in sialylation. However, a few clarifications and some softening of mechanistic and translational claims are needed to ensure the conclusions are fully supported by the data presented and the work is readily reproducible. Please see my specific comments below:

-it would be of value to strengthen the reporting of biological replicates and donor handling, which is key for reproducibility. The endothelial assays use HUVECs and HCMECs from two donors each, which is a good start, but it is not always clear whether n refers to independent EMC isolations/conditioned media preparations, technical repeats, endothelial donors or wells/images. Please state explicitly for each main figure (i) the number of independent epi-condm preparations as biological replicates, (ii) whether analyses were performed per donor and then pooled or otherwise and (iii) how donor-to-donor variability was handled statistically.

Response: We thank the reviewer for emphasizing the importance of transparent reporting of biological replication. We have revised the Methods section and figure legends to clearly describe the experimental hierarchy and handling of replicates.

Specifically:

- For each experimental condition (normoxia, NM; hypoxia, HP; and reoxygenation, RO), epicardial conditioned media (Epi-CondM) were generated from independent epicardial mesothelial cells (EMC) cultures from individual passages. Each dish represents an independent biological replicate at the level of EMC culture and secretome production. **Clarification is now found in Methods Section (pg 13, ln 402-404).**
- Endothelial functional assays were performed by applying the same Epi-CondM preparation to HUVECs and HCMECs derived from two independent donors each. Thus, each Epi-CondM biological replicate was tested across both endothelial donors.
- For quantitative analyses, readouts obtained from the two donors were averaged to generate a single value per Epi-CondM preparation, thereby preserving the Epi-CondM as the primary biological unit of analysis. The data presented in the main figures therefore reflect the effects of independent Epi-CondM preparations on endothelial function, which allows to assess inter-donor variability without over-weighting endothelial donor effects in the primary statistical comparisons.

-while the choice of epicardial mesothelial cells and both HUVECs and HCMECs is widely used in the field, the rationale for selecting these specific in vitro models is largely implicit. We would welcome a brief justification of their relevance and limitations to further strengthen the experimental framework.

Response: We have added a brief justification in the Result sections, **which reads as follows:**

EMCs were selected as they represent a widely used in vitro surrogate for the adult epicardium and allow controlled interrogation of epicardial paracrine outputs (Wei et al., 2015) (pg 4, ln 86-88). HUVECs and HCMECs were included to capture both a standardized endothelial model and a cardiac-specific microvascular context, respectively (pg 5, ln 130-131). We acknowledge the limitations of these simplified in vitro systems.

-the model relies on very late passage "primary", where passages 25-30 were used. This is unusually high for primary cells and could materially affect secretory phenotype and stress responses. Please justify and/or control for this- at minimum, please justify why this passage range is appropriate and consider adding a sensitivity analysis, e.g. repeating a key experiment with earlier passages or

independent

isolations.

Response: EMCs cells were characterised as epicardial cells with highly expressed WT1 and other epicardial markers (shown in revised Fig S1).

A comparative analysis to earlier passages yielded highly consistent results, demonstrating a stable secretory phenotype. These are:

1. High Qualitative Overlap: Despite the differences in MS platforms, 87.6% (263 out of 300) of the secreted proteins identified in the Wei et al dataset were robustly detected in the current study's dataset.

2. Quantitative comparison of the core secretome (n = 263 overlapping proteins) between the two independent isolations/batches revealed a highly significant positive correlation in relative protein abundances (Spearman's $r = 0.5726$, 95% CI: 0.4825 to 0.6507, $P < 0.0001$). This robust correlation, achieved despite utilizing distinct MS platforms and analytical metrics, definitively demonstrates that the primary cells at passages 25-30 maintain a remarkably stable and consistent secretory phenotype, without shifting towards an aberrant or stress-induced secretome profile.

3. Preserved Core Secretome: The dominant functional secreted proteins remained identical between the two independent isolations. Both datasets were highly enriched in key extracellular matrix components, with Col1a1, Col1a2, Col3a1, Col5a2, Bgn (Biglycan) and Igfbp2 being the most abundantly secreted proteins. We did not observe a shift towards a senescence-associated secretory phenotype or an abnormal domination of stress-response proteins in the P25-30 group.

-the results state that the loss of pro-proliferative activity persists after reoxygenation (HP and RO indistinguishable in viability assays, Fig 1D-E), yet tube formation shows partial restoration in HCMECs with RO Epi-CondM (Fig 3G-J). The lectin section suggests restored $\alpha 2,6$ -sialylation "might explain" the recovered angiogenic activity. These are not contradictory, but the manuscript should more clearly separate (i) proliferation/viability effects and (ii) network formation effects. This would avoid implying a single unified "recovery" phenotype without specifying which functional readout.

Response: We agree that clearer separation of functional readouts improves interpretability. We have revised the Results and Discussion to explicitly distinguish between:

- Endothelial viability/proliferation, which remains suppressed after reoxygenation, and
- Network formation/angiogenic organization, which shows partial recovery in HCMECs.

We now avoid describing a single unified "recovery" phenotype and instead emphasize that reoxygenation selectively restores certain angiogenic properties without fully normalizing endothelial survival responses.

-it is a particular strength of the manuscript that the authors include a clear and balanced "limitations" section, which appropriately contextualises the in vitro nature of the work and avoids overstatement of translational impact. Indeed, the suggested mechanistic causality is still incomplete as acknowledged by the authors in this section - while LOF evidence is supportive of this, causal inference would be stronger with a rescue set up. Therefore, we would ask the authors to temper claims and/or add a rescue experiment on the conclusion that ST6GAL1-dependent $\alpha 2,6$ -sialylation is required revealing a new therapeutic target. We would recommend softening the abstract and discussion language to "suggests/consistent with/may represent" unless a rescue is added. Related to this, the "therapeutic target" and in vivo relevance are currently speculative given all findings are in vitro. We recommend toning down statements implying translational readiness in the abstract and conclusions and framing therapeutic relevance as a hypothesis for future testing.

Response: We thank the reviewer for this thoughtful comment and agree that causal inference should be framed conservatively. In line with this recommendation, we have softened mechanistic and translational language throughout the manuscript, including the Abstract, Results, and Discussion.

All statements implying requirement or therapeutic targeting of ST6GAL1-dependent $\alpha 2,6$ -sialylation have been revised to language such as "suggests," "is consistent with," or "may

contribute to.” We now clearly state that while loss-of-function data support a functional role, definitive causality would require rescue-based approaches, which are beyond the scope of the current study. Therapeutic relevance is explicitly framed as a hypothesis for future investigation rather than a demonstrated application.

-please clarify the hypoxia monitoring replication and sampling strategy. Oxygen level monitoring uses a randomly selected culture dish from each group. This reads as n=1 dish per condition/timepoint, which is not ideal. Please report how many independent dishes/experiments were quantified for hypoxia reagent imaging and for EMC viability (Fig S2), and whether the "random dish" approach was repeated across independent runs.

Response: Hypoxia monitoring using oxygen-sensitive probes was performed across multiple independent experiments, with a randomly selected dish imaged per condition per experimental run (passage), not as a single-dish measurement overall. EMC viability assays were similarly replicated across independent cultures. We have now clarified the number of independent experiments and dishes analyzed, which can be found in the Methods section (pg 14, ln 448).

-the transcriptomic re-analysis is based on ArrayExpress E-MTAB-15094, please provide sufficient parameters for re-analysis such as group definitions, threshold and any batch handling that was performed during the analysis.

Response: We have now expanded the Methods section to include: explicit group definitions used for re-analysis of ArrayExpress E-MTAB-15094; normalization procedures and differential expression analysis (pg 17, ln 526-537). This information should enable full reproducibility of the transcriptomic analysis.

Reviewer 2: The authors' data supports some of the conclusions, but given the limitations of the study, there is insufficient evidence that this mechanism plays any role in the epicardial response upon ischemic injury.

This model relies entirely on in vitro data, therefore drawing conclusions about the actual role of epicardium signalling in ischemic (or IR) injury is very unlikely to be reflected. Moreover, the authors combine two different model organisms and do not address whether this response is maintained across species (and very different stages of development, as is the case between this epicardial cell line and HUVECs).

Response: We appreciate the reviewer's critical perspective and agree that in vitro findings cannot directly define in vivo epicardial behavior during ischemic or ischemia-reperfusion injury. We have therefore revised the manuscript to more clearly state that our conclusions are limited to controlled in vitro modeling of epicardial secretory responses to severe hypoxia. All language implying a direct role in ischemic injury has been softened, and in vivo relevance is now explicitly framed as speculative and hypothesis-generating rather than demonstrative.

The epicardial cell line used presents specific caveats: it is a dormant epicardium isolated from adult mouse heart. Therefore, it has been immortalized in an epithelial state. It is known that the epicardium undergoes EMT and becomes activated, both in development and in response to injury and it is this specific phenotype that presents the most relevant paracrine signalling. Moreover, it is highly unlikely that these severe hypoxic levels would be reached (even in extreme ischemic injury) in the outermost layer of the heart (where epicardial cells in epithelial state reside).

Response: We agree that the epicardium is a dynamic tissue that undergoes EMT and activation in development and injury. The EMC model used here represents a quiescent epithelial epicardial state and was intentionally chosen to isolate how extreme oxygen deprivation alone alters baseline epicardial paracrine output, independent of EMT or inflammatory cues.

We now clarify this rationale in the Discussion and explicitly acknowledge that activated or EMT-transitioned epicardial cells may exhibit distinct secretory and glycosylation responses. Similarly, we acknowledge that 0.1% O₂ likely represents an extreme condition and is best interpreted as a stress-testing paradigm rather than a physiological oxygen level experienced uniformly by epicardial cells in vivo.

In order to address this major issue, the ideal scenario would be to address epicardial secretome in vivo after hypoxic conditions/ischemic damage and whether sialylation is also taking place in a significant manner (alternatively, Stgal1 levels could be evaluated).

Alternatively, addressing if this response is conserved, primary epicardial cells from mouse or human model could be cultured and their Epi-CM evaluated for PTM under severe hypoxic conditions.

Response: We fully agree that in vivo validation and cross-species comparisons would substantially strengthen mechanistic conclusions. However, these directions are beyond the scope of the present study. We have now explicitly included these points in the Limitations section and frame them as important future directions, including in vivo assessment of epicardial sialylation and ST6GAL1 regulation following ischemic injury.

Another point that should be addressed how the diminishing of cell viability upon extreme hypoxia of epicardial cells is behind the response and changes in CM. Some other stress-inducing affecting variability could also be tested.

If primary epicardial cultures are evaluated, how does extreme hypoxia alter specifically the activation phenotype of epicardial cells and does that impact its secretome?

Response: We thank the reviewer for raising this important point regarding the impact of reduced epicardial cell viability and hypoxia-induced stress on conditioned media composition. In addition to protein normalization of all Epi-CondM preparations, we have performed an unbiased proteomic analysis of Epi-CondM collected under normoxia, hypoxia, and reoxygenation conditions.

This proteomic profiling enables a systematic assessment of qualitative changes in the epicardial secretome under extreme hypoxic stress, beyond total protein output. Importantly, all downstream endothelial functional assays were conducted using protein-normalized Epi-CondM, ensuring that differences in endothelial responses are not attributable to reduced cell number or bulk protein loss, but instead reflect stress-associated remodeling of secreted protein composition.

We acknowledge that extreme hypoxia represents a strong stress stimulus and may induce broad adaptive or maladaptive secretory responses. However, the combination of protein normalization and secretome-level proteomic analysis allows a controlled and interpretable comparison of Epi-CondM bioactivity across oxygen conditions. We have revised the Methods, Results, and Discussion sections to clarify this experimental design and its implications.

Regarding the reviewer's question on epicardial activation phenotypes, we now explicitly state that the current study focuses on secretome alterations in a defined epicardial mesothelial cell line and does not address hypoxia-induced EMT or activation states of primary epicardial cells, which represent important directions for future investigation.

Lastly, the authors evaluate transcription of proangiogenic factors (Such as VEGFa and c, and Fgf2) upon severe hypoxia of epicardial cells however the proangiogenic response is abrogated. However, the presence of these factors in the conditioned media is not evaluated.

Response:. To directly address this, we supplemented our analysis with proteomic profiling of epicardial conditioned media (Epi-CondM), as indicated in Fig. 1D and page 17-18.

The proteomics data reveal that several canonical angiogenic regulators, including VEGFA, VEGFC, THBS1, and SERPINF1, are readily detectable in Epi-CondM and exhibit condition-dependent abundance changes across NM, HP, and RO states. Notably, both pro-angiogenic (e.g., VEGFA, VEGFC) and anti-angiogenic or context-dependent modulators (e.g., THBS1, SERPINF1) are dynamically regulated, indicating that endothelial functional outcomes cannot be inferred from individual factor levels alone.

Accordingly, we have revised the manuscript to emphasize that the impaired angiogenic activity of hypoxic Epi-CondM reflects the integrated effect of secretome rebalancing, rather than the

absence of specific pro-angiogenic proteins. This interpretation is now supported by direct protein-level evidence and is consistent with the functional endothelial assays presented.

Reference

Wei, K., Serpooshan, V., Hurtado, C., Diez-Cuñado, M., Zhao, M., Maruyama, S., Zhu, W., Fajardo, G., Nosedá, M., Nakamura, K., et al. (2015). Epicardial FSTL1 reconstitution regenerates the adult mammalian heart. *Nature* 525, 479-485.

Second decision letter

MS ID#: bio.062479R1

MS Title: Severe hypoxia drives loss of ST6GAL1-mediated α 2,6-sialylation in the epicardial secretome impairing angiogenic activity

Authors: Pilar Ruiz-Lozano, Junqing Zhang, Harry J. Whitwell and Costanza Emanuelli

I am happy to tell you that your manuscript has been accepted for publication in Biology Open, pending our standard publication integrity checks. It was accepted on 30th March 2026.